# A Brain Morphometry Study with Across-Site Harmonization Using a ComBat-Generalized Additive Model in Children and Adolescents

**DOI:** 10.3390/diagnostics13172774

**Published:** 2023-08-27

**Authors:** Tadashi Shiohama, Norihide Maikusa, Masahiro Kawaguchi, Jun Natsume, Yoshiyuki Hirano, Keito Saito, Jun-ichi Takanashi, Jacob Levman, Emi Takahashi, Koji Matsumoto, Hajime Yokota, Shinya Hattori, Keita Tsujimura, Daisuke Sawada, Tomoko Uchida, Tomozumi Takatani, Katsunori Fujii, Shinji Naganawa, Noriko Sato, Hiromichi Hamada

**Affiliations:** 1Department of Pediatrics, Graduate School of Medicine, Chiba University, Inohana 1-8-1, Chuo-ku, Chiba-shi 260-8677, Chiba, Japan; 2Division of Newborn Medicine, Department of Medicine, Boston Children’s Hospital, Harvard Medical School, 300 Longwood Avenue, Boston, MA 02115, USA; 3Center for Evolutionary Cognitive Sciences, Graduate School of Art and Sciences, The University of Tokyo, Tokyo 108-8639, Japan; 4Department of Radiology, National Center Hospital, National Center of Neurology and Psychiatry, Tokyo 187-8551, Japan; 5Department of Pediatrics, Nagoya University Graduate School of Medicine, 65 Tsurumai-cho, Showa-ku, Nagoya 466-8550, Aichi, Japan; doctormasa2011@gmail.com (M.K.);; 6Department of Developmental Disability Medicine, Nagoya University Graduate School of Medicine, 65 Tsurumai-cho, Showa-ku, Nagoya 466-8550, Aichi, Japan; 7Research Center for Child Mental Development, Chiba University, Inohana 1-8-1, Chuo-ku, Chiba-shi 260-8677, Chiba, Japan; 8United Graduate School of Child Development, Osaka University, Kanazawa University, Hamamatsu University School of Medicine, Chiba University and University of Fukui, Suita 565-0871, Osaka, Japan; 9Department of Pediatrics and Pediatric Neurology, Tokyo Women’s Medical University Yachiyo Medical Center, 477-96 Owadashinden, Yachiyo-shi 276-8524, Chiba, Japan; 10Department of Mathematics, Statistics and Computer Science, St. Francis Xavier University, 5005 Chapel Square, Antigonish, NS B2G 2W5, Canada; 11Athinoula A. Martinos Center for Biomedical Imaging, Department of Radiology, Massachusetts General Hospital, Harvard Medical School, 149 13th Street, Charlestown, MA 02129, USA; 12Nova Scotia Health Authority—Research, Innovation and Discovery Center for Clinical Research, 5790 University Avenue, Halifax, NS B3H 1V7, Canada; 13Department of Radiology, Chiba University Hospital, Inohana 1-8-1, Chuo-ku, Chiba-shi 260-8677, Chiba, Japan; 14Diagnostic Radiology and Radiation Oncology, Graduate School of Medicine, Chiba University, Inohana 1-8-1, Chuo-ku, Chiba-shi 260-8677, Chiba, Japan; 15Group of Brain Function and Development, Neuroscience Institute of the Graduate School of Science, Nagoya University, Nagoya 466-8550, Aichi, Japan; 16Research Unit for Developmental Disorders, Institute for Advanced Research, Nagoya University, Nagoya 466-8550, Aichi, Japan; 17Department of Pediatrics, International University of Welfare and Health School of Medicine, Narita 286-8520, Chiba, Japan; 18Department of Radiology, Nagoya University Graduate School of Medicine, 65 Tsurumai-cho, Showa-ku, Nagoya 466-8550, Aichi, Japan

**Keywords:** ComBat-GAM, structural brain MRI, voxel-based morphometry, CIVET, normal reference values

## Abstract

Regional anatomical structures of the brain are intimately connected to functions corresponding to specific regions and the temporospatial pattern of genetic expression and their functions from the fetal period to old age. Therefore, quantitative brain morphometry has often been employed in neuroscience investigations, while controlling for the scanner effect of the scanner is a critical issue for ensuring accuracy in brain morphometric studies of rare orphan diseases due to the lack of normal reference values available for multicenter studies. This study aimed to provide across-site normal reference values of global and regional brain volumes for each sex and age group in children and adolescents. We collected magnetic resonance imaging (MRI) examinations of 846 neurotypical participants aged 6.0–17.9 years (339 male and 507 female participants) from 5 institutions comprising healthy volunteers or neurotypical patients without neurological disorders, neuropsychological disorders, or epilepsy. Regional-based analysis using the CIVET 2.1.0. pipeline provided regional brain volumes, and the measurements were across-site combined using ComBat-GAM harmonization. The normal reference values of global and regional brain volumes and lateral indices in our study could be helpful for evaluating the characteristics of the brain morphology of each individual in a clinical setting and investigating the brain morphology of ultra-rare diseases.

## 1. Introduction

The regional anatomical structures of the brain are intimately connected to functions corresponding to specific regions and the temporospatial pattern of genetic expression from the fetal period to old age [1,2,3,4]. Structural features, such as regional cortical thicknesses and curvatures, are not just beneficial in characterizing morphology but are also tightly associated with the brain functions corresponding to their location. Indeed, many studies using brain magnetic resonance imaging (MRI) have provided strong evidence that brain morphology is associated with intelligence [5,6] and mental and developmental disorders [7,8]. Brain MRI is recognized as one of the most useful non-invasive modalities with high spatial resolution in three dimensions for evaluating brain morphology in living participants. Thus, brain quantitative morphometric approaches using brain anatomical structural MRI examinations have often been employed in neuroscience investigations of neurotypical development [9,10,11,12,13,14], mental and developmental disorders, including schizophrenia [15,16], major depression [15,17], autism spectrum disorder [15,18], and attention-deficit/hyperactivity disorder [19,20]. However, except for relatively common diseases, such as Down syndrome [21,22] and Rett syndrome [23,24,25], brain morphology in most congenital genetic disorders has not been studied to date because it is difficult to collect many examples of the same rare disease in a single facility.

Group comparisons of measures calculated from three-dimensional T1-weighted images between affected and non-affected participants have been widely used to reveal group-wise differences in brain morphology. Importantly, we should carefully attempt to control for major covariates, such as differences in sex, age, and comorbidities. In addition to these major covariates, measurement bias due to differences in MRI scanners should not be overlooked when evaluating brain morphology [26,27,28,29]. For example, differences among MRI scanners have been reported to generate 0.59 standard deviation changes in cortical volume in voxel-based morphometry [28] and 0.4 mm changes in cortical thickness in surface-based morphometry [29]. Additionally, it is not realistic that patients with rare diseases assemble at a single-scan site because the rarity of their condition makes it highly likely that they live scattered over a wide area. Therefore, it is difficult to conduct multicenter studies of brain morphology in ultra-rare disorders.

Brain structural changes in a pediatric population were evaluated in several studies [9,10,11,12,13,14], while there are a few reports providing normal references of brain structural measurements for each age group [9]. To our knowledge, there are no normal references that can be generalized across multiple facilities.

In this study, we proposed a novel approach to evaluate the morphological characteristics of the brain in children by expanding the indications of ComBat-generalized additive model (GAM) harmonization [30,31]. Although this harmonization model was originally designed to be employed in international datasets for specific disorders, our approach is useful for evaluating brain morphology in a wide range of cases. The results of our study provide normal reference values for global and regional brain volumes for each sex and age group. The normal reference values in our study would be helpful for evaluating the characteristics of the brain morphology of each individual in a clinical setting and for investigating the brain morphology of ultra-rare diseases.

## 2. Materials and Methods

### 2.1. Participants and MRI Acquisition

We obtained study approval from the local institutional review boards of five institutions: Boston Children’s Hospital (BCH), Department of Pediatrics, Research Center for Child Mental Development, Chiba University (CHBC), Chiba University Hospital (CUH), Nagoya University Hospital (NGO), and Tokyo Women’s Medical University Yachiyo Medical Center (TWYM). The MRI examinations of neurotypical participants (NPs) aged 6.0–17.9 years old were collected from the in-house database of each institution comprising healthy volunteers or neurotypical patients without neurological disorders, neuropsychological disorders, or epilepsy. Healthy volunteers were typically developing children recruited from a population who did not need physician visits, while neurotypical participants consisted of clinical patients who were determined to not have any neurological disorders, and for whom a neuroradiologist assessed their brain MRI examination as normal. Written informed consent was obtained from the guardians of the healthy volunteers from the centers in Japan. Informed consent was waived by BCH’s institutional review board for retrospective analysis.

Three-dimensional (3-D) T1-weighted images were obtained with clinical MRI scanners in BCH (MAGNETOM Skyra 3.0T, Siemens Medical Systems, Erlangen, Germany), CHBC (Discovery MR750 3.0T, GE Healthcare, Milwaukee, WI, USA), CUH (GE Signa HDxT 1.5T, GE Healthcare, Milwaukee, WI, USA), NGO (Magnetom Prisma 3.0T, Siemens Medical Systems, Erlangen, Germany), and TWYM (Ingenia 3.0T CX, Philips Healthcare, Best, The Netherlands). An MRI was performed on participants under natural sleep or sedation. The acquisition setting parameters, such as repetition time, echo time, inversion time, voxel size, and matrix size, are listed in Table 1. The quality of the acquired images was visually evaluated, and poor-quality images were excluded from the analysis despite motion correction.

### 2.2. Structural MRI Processing

DICOM files of 3D-T1-weighted images were converted to anonymous NIfTI files using dcm2nii from the MRIcron software package v1.0.2 [32] and analyzed with the CIVET version 2.1.0 pipeline [33] on the CBRAIN platform [34]. Corrections for non-uniform intensity artifacts by the N3 algorithm [35], stereotaxic registration onto the icbm152 nonlinear 2009 template [36], and brain masking [37] were performed. A region-based volumetric analysis was performed with tissue classification using an artificial neural network classifier (INSECT) [38]; the segmentation of brain regions was performed with ANIMAL [39], and 36 regional volumes were calculated for each image. The quality of the outputs of the CIVET pipeline (brain mask shapes, linear/nonlinear registration to the template, tissue classification, and brain segmentation) was visually inspected (Figure 1a). The regional and global volumes are listed in Table 2. Global brain volumes were calculated from regional volumes corrected using ComBat-GAM. The mean and standard deviation in each regional brain volume were analyzed for each sex and age group of 6.0–8.9, 9.0–11.9, 12.0–14.9, and 15.0–17.9 years.

### 2.3. ComBat-GAM Harmonization and Attempting the Trained ComBat-GAM Model

The ComBat-GAM program (https://github.com/rpomponio/neuroHarmonize/, accessed on 1 February 2023) was executed using Python version 3.8.5 scripts. Scatter plots of regional volumes of the right parietal gray matter (Figure 1b) and other regions (Appendix A) for scan age and scan site before and after ComBat-GAM harmonization were visually evaluated to confirm the control of the scan-site effect using R for Windows 4.2.2 and Rstudio 2023.03.0 + 386 (https://posit.co/download/rstudio-desktop/, accessed on 1 February 2023). By excluding measurement bias using ComBat-GAM harmonization with 846 NPs in our cohort, we corrected the values of regional brain volumes in 18 additional cases and evaluated the validity of this pipeline, i.e., we trained a harmonization model using the “harmonizationLearn” function with 846 NPs and applied the model parameters for controlling measurement bias in 846 NPs and 18 additional cases using the “harmonizationApply” function.

### 2.4. Laterality Index

We also investigated brain volume laterality using the laterality index (LI). The LI was calculated for each regional volume as the ratio [VL − VR]/[VL + VR] × 100 (where VL and VR are regional volumes for the left and right hemispheres, respectively) [40]. LIs were subsequently classified as left hemisphere dominant (defined as LI > 20), symmetric (−20 ≤ LI ≤ +20), or right hemisphere dominant (LI < −20) [40].

### 2.5. Statistical Analyses

IBM SPSS Statistics for Windows version 28 (IBM Corp. Armonk, NY, USA) was used to visualize the normal reference values and age transitions. Z-standardized scores were obtained from a radar chart using Microsoft Excel 2019 (Redmond, WA, USA). Each brain structure measurement and the laterality index in male and female participants were evaluated through Welch’s two-tailed unpaired *t*-tests with the Bonferroni correction for multiple comparisons and the absolute value of Cohen’s *d* statistic. We determined *p* < 1.4 × 10^−3^ and *p* < 3.1 × 10^−3^ (α = 0.05 for 42 and 16 repeating *t*-tests) as a statistically significance level for the structural measurements and laterality indices, respectively. Cohen’s *d* = 0.8 was recognized as the cut-off value for large-size effects [41].

## 3. Results

### 3.1. Participants’ Background

Our retrospective study included 846 images from 846 NPs aged 6.0–17.9 years after excluding one image with segmentation failure on the CIVET pipeline. The mean (standard deviation (SD)) age at MRI examination was 11.7 (3.3) and 12.8 (3.4) years in males (*n* = 339) and females (*n* = 507), respectively. We examined images of an additional 18 cases obtained from Japanese sites to reconfirm the usefulness of the trained ComBat-GAM model, which was established by analyzing 846 normal controls. The clinical information and scan site for each additional case are listed in Appendix A. Additional cases included three patients with Gorlin syndrome, one with Malan syndrome, four with PTEN hamartoma tumor syndrome, three neurotypical controls, and seven with Rett syndrome (Appendix A).

### 3.2. Global and Regional Brain Volumes and Lateralities

The 36 regional brain volumes were measured for each image using the CIVET pipeline. The mean (SD) of global brain volumes was calculated from regional brain volumes corrected using ComBat-GAM as follows: 1902 (163) and 1747 (138) cm^3^ of the whole brain (WB); 778 (71) and 692 (80) cm^3^ of the cerebral gray matter and fornix (CGM); 481 (62) and 432 (51) cm^3^ of the white matter (WM); 38 (3) and 36 (3) cm^3^ of the subcortical gray matter and fornix (SGM); 419 (90) and 416 (83) cm^3^ of the cerebral spinal fluid space (CSF); and 185 (15) and 170 (14) cm^3^ in the cerebellum and brainstem (CB) for males and females, respectively. For each sex and age group of 6.0–8.9, 9.0–11.9, 12.0–14.9, and 15.0–17.9 years, the mean and SD of global and regional brain volumes are provided in Table 3 and Appendix A.

In terms of sex-based differences in global and regional brain volumes, almost all measurements except for the cerebrospinal fluid system have significantly larger volumes in male participants compared to female participants in each age group (Table 3 and Appendix A).

In terms of global and regional brain volumes (Figure 2), the regional volume of the CSF increased remarkably, and the WB, WM, and CB increased slightly with advancing age. Scatter plots showed the regional volumes of each region of the CGM and WB (Appendix A), SGM (Appendix A), and CSF and CB (Appendix A) for each sex and age group.

The LI of regional brain volumes consistently attained a value of approximately zero with no connection to age in most brain regions, whereas the LI values in the occipital GM, occipital WM, subthalamic nucleus, and lateral ventricle were widely scattered (Table 4 and Figure 3). There was no significant difference in the LI between male and female participants (Table 4).

### 3.3. Applying the Trained ComBat-GAM Model to Additional Cases

To reconfirm the usefulness of the trained ComBat-GAM model, which was established by analyzing 846 normal controls, we applied the model to an additional 18 cases to provide Z-scores for global brain volumes (Figure 4 and Appendix A), regional brain volumes (Appendix A), and the LI (Appendix A) for each case. In patients with Gorlin syndrome (Cases 1–3), unlike WM and CSF, CGM, SGM, and CB were 2 SD larger than the normal reference values. In patients with Malan syndrome (Case 4) or PTEN hamartoma tumor syndrome (Cases 5–9), WM, CSF, CGM, and SGM were larger than the normal reference values (Figure 4). In patients with Rett syndrome (Cases 13–18), only patients aged >7 years (Cases 16–18) showed decreased volumes of CGM, SGM, and CB (Figure 4). Almost all cases exhibited symmetric patterns, except for asymmetry in the lateral ventricle in three cases (Cases 2, 4, and 8) (Appendix A).

## 4. Discussion

This study provided normal reference values of regional brain volumes for each sex and age group through across-site harmonization using the ComBat-GAM in children and adolescents. The trained ComBat-GAM model successfully assessed the brain morphological characteristics of additional cases of normal controls and patients with congenital diseases.

ComBat was originally developed to control batch effects on multiple gene arrays [42] and was subsequently adapted to control site effects in neuroimaging studies [29]. The most common approach for statistically combining MRI datasets with site covariates is the high-level general linear model implemented to assess group differences [43,44,45]; however, the nonlinearity of age trends of various brain regional volumes at a young age has been observed in previous reports [10,11,14,30], unlike the adult population.

ComBat-GAM [30,31] is an extension of ComBat harmonization with a GAM to adjust for nonlinear associations between covariates (such as sex, age, and scan site) and image summaries (such as regional volumes and curvature measurements). Another major approach, the linear mixed-effects modeling (LME) method, has been proposed [46]. In healthy participants ranging in age from 3 to 96 years, when compared to ComBat [30,31] or LME [31], ComBat-GAM has an advantage in the across-site harmonization of regional brain volumes with age-related nonlinearities. We employed ComBat-GAM in our study, which consisted of participants aged 6.0–17.9 years.

Most brain morphometry programs, such as the CIVET pipeline, are optimized for brain MRI of participants aged >6 years; therefore, we focused on providing the normal reference values of participants aged 6.0–17.9 years. Our data confirmed that the volume of CGM decreased after approximately 6 years of age during adolescence and WM continuously increased, as previously reported [10,11,14,47]. In terms of sex-based effects, statistically significant differences were observed in almost all brain global and regional volumes, except for CSF systems.

We provided normal reference values for six global volumes and thirty-six regional volumes for each sex and age group. Through ComBat-GAM harmonization in the current study, brain 3D-T1-weighted images scanned at each institution could be individually assessed as Z-standardized scores, contributing to the advancement of brain morphology evaluation in a clinical setting.

We also showed symmetry as a laterality index of the regional brain volumes. Previous studies have demonstrated regional asymmetry in cortical volume [9,48], surface area [9,49], cortical thickness [9,49], and subcortical brain regional volume [50] as disease-related changes. This study successfully presented regional asymmetry connected with sex, age, and heritability in healthy populations, while at the same time, the brain laterality was shown to generally vary within the normal symmetrical range (−20 ≤ LI ≤ +20), as shown in our results. The LI showed no significant sex-based differences. In contrast, brain asymmetry was observed in several congenital genetic disorders, such as CHARGE syndrome [51], megalencephaly capillary malformation [52], Aicardi syndrome [53], fetal alcohol syndrome [54], and Sturge–Weber syndrome [55]. Although these studies were mainly conducted through qualitative rather than quantitative analyses, our data on the LI may contribute to the quantitative assessment of regional brain symmetry in children and adolescents with some disorders.

### Limitations and Future Directions

In the current study, we trained a ComBat-GAM model from only neurotypical participants (healthy volunteers and neurotypical participants), and subsequently tested the model in another population with or without congenital disorders. It remains unclear whether to analyze only NPs or both NPs and abnormal cases, but the trained ComBat-GAM model could accurately identify abnormalities in regional brain volumes for each sex and age group. We cannot completely exclude the possibility that our approach using the trained ComBat-GAM model had low relevance when the distribution of covariates between the NPs and abnormal cases was largely different. Even after considering this limitation, we prioritized handling additional individual cases with various diseases by building a trained ComBat-GAM model by analyzing only NPs.

Our study focused only on regional brain volumes. Quantitative measurements of structural brain morphology include surface area, cortical thickness, and curvature measurements (for quantifying local gyral and sulcal structures) [56,57]. Regional brain volumes are also widely employed in brain morphological studies. Another limitation is that we could not distinguish regional, scan machine, and racial effects on regional brain volumes because only one institution (BCH) is located in the USA and the other institutions are located in Japan.

As a future direction, a similar study with a larger patient population could be conducted to increase accuracy. Furthermore, the across-site harmonization using ComBat-GAM in this study can be applied to normalization in the structural brain morphology in infantile participants, as well as in studies focused on fiber volumes derived from diffusion tensor imaging-based tractography.

## 5. Conclusions

We provided the across-site normal reference values of global and regional brain volumes for each sex and age group in children and adolescents through ComBat-GAM harmonization in 846 NPs. The reference in our study would be helpful for evaluating the characteristics of the brain morphology of each individual in a clinical setting and investigating the brain morphology of ultra-rare diseases.

## Figures and Tables

**Figure 1 diagnostics-13-02774-f001:**
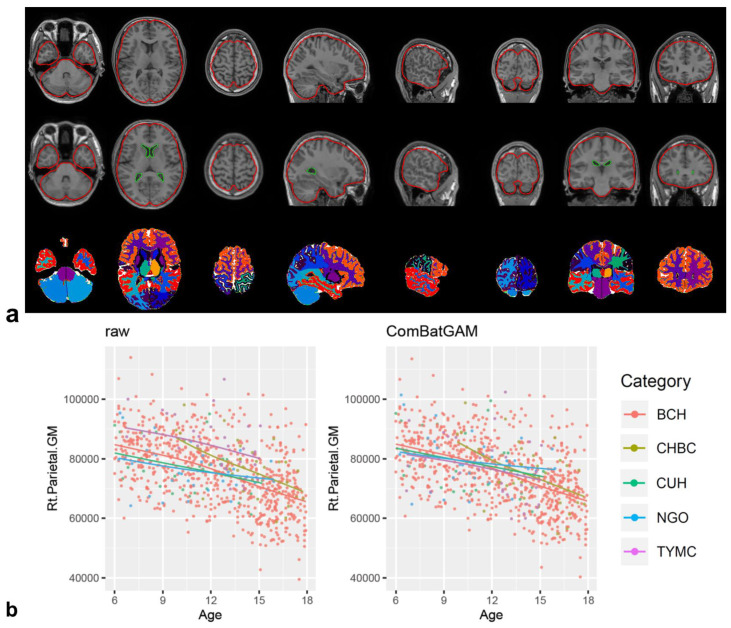
CIVET pipeline and ComBat-GAM harmonization. Visualization of the registration quality of the CIVET pipeline showed linear registration, nonlinear transformation to the stereotaxic model, and ANIMAL segmentation from top to bottom (**a**). Scatter plot of the regional brain volume of the right parietal gray matter according to scan age and scan site before ((**b**), left panel) and after ComBat-GAM harmonization ((**b**), right panel). Each color of dot and line indicates scan-site as shown in the category and Table 1.

**Figure 2 diagnostics-13-02774-f002:**
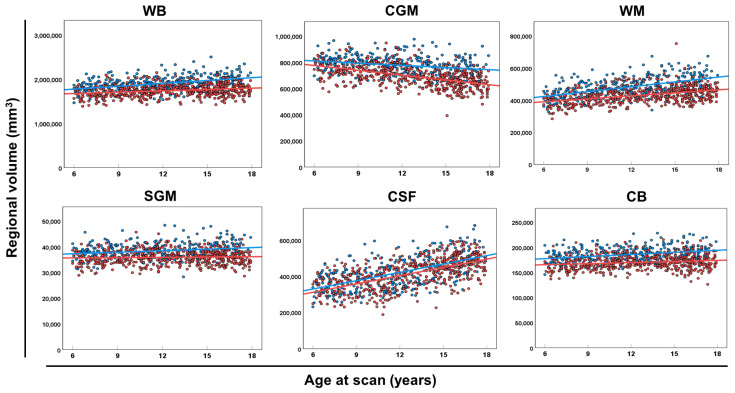
The global volume of each brain category. Scatter plots and regression lines (between age at the scan and regional volume of WB, CGM, WM, SGM, CSF, and CB) in male (blue circles and lines) and female (red circles and lines) neurotypical controls. Abbreviations: CB, cerebellum and brainstem; CGM, cerebral gray matter; CSF, extra-axial cerebrospinal fluid; SGM, subcortical gray matter and fornix; WB, whole brain; WM, white matter.

**Figure 3 diagnostics-13-02774-f003:**
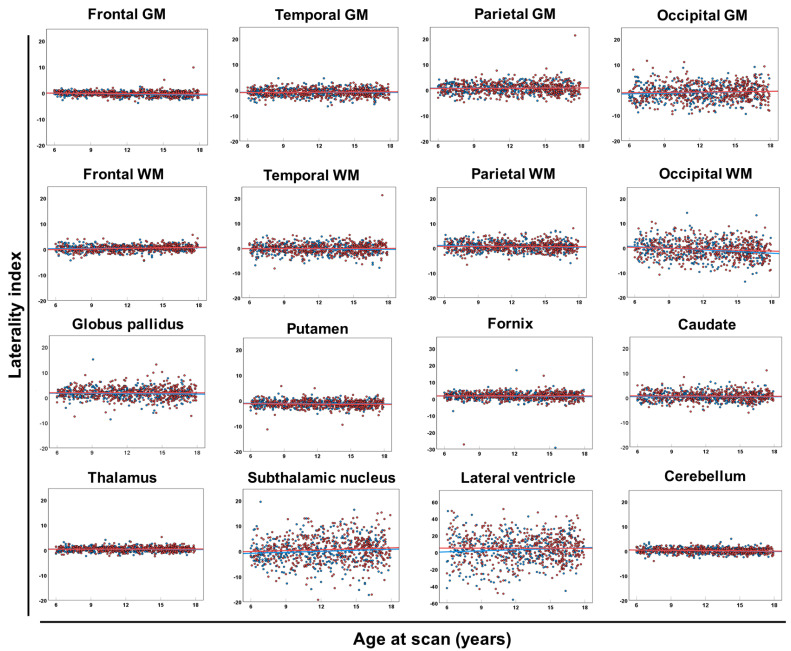
Laterality index of each brain region. Scatter plots and regression lines (between age at scan and laterality index of each brain region) in male (blue circles and lines) and female (red circles and lines) neurotypical controls. Abbreviations: GM, cerebral gray matter; WM, white matter.

**Figure 4 diagnostics-13-02774-f004:**
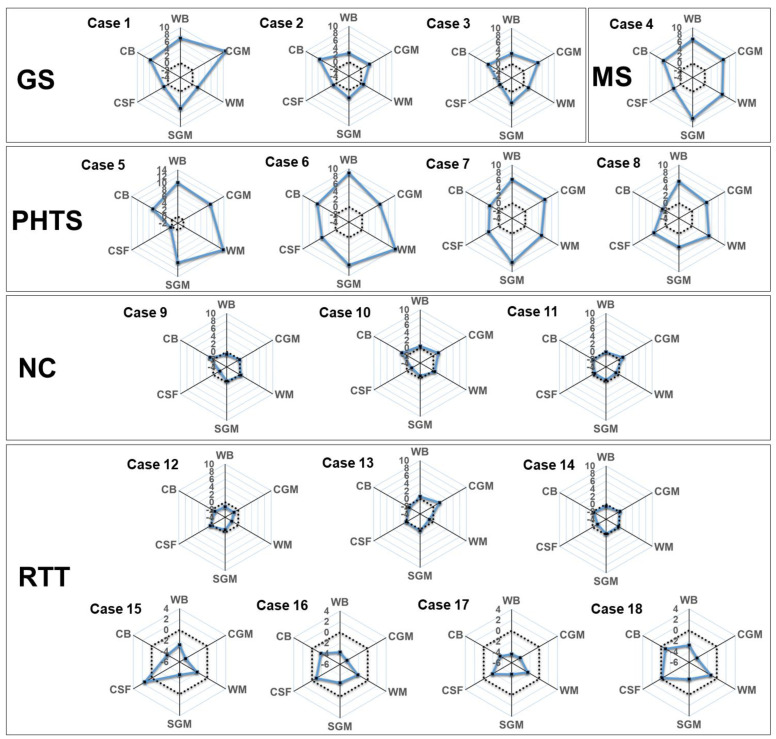
Rader charts for the global brain volumes in each additional case. Z-standardized scores according to our normal reference values of WB, CGM, WM, SGM, CSF, and CB are plotted (black circles and solid blue lines) with mean level (dot line). Abbreviations: CB, cerebellum and brainstem; CSF, cerebrospinal fluid system; CGM, cerebral gray matter; GS, Gorlin syndrome; NC, neurotypical control; PHTS, PTEN hamartoma tumor syndrome; RTT, Rett syndrome; SGM, subcortical gray matter and fornix; MS, Malan syndrome; WB, whole brain; WM, white matter.

**Table 1 diagnostics-13-02774-t001:** Background and MRI scanner settings in each cohort.

Scan-Site	BCH	CHBC	CUH	NGO	TYMC
Subject	Neurotypical controls	Healthy volunteers	Neurotypical controls	Healthy volunteers	Neurotypical controls
Sex (*n*)	Male 274 Female 438	Male 20 Female 16	Male 14 Female 16	Male 19 Female 15	Male 12 Female 22
Age at scan (mean (SD), years)	12.4 (3.4)	14.2 (2.1)	10.8 (2.8)	10.9 (2.9)	12.0 (2.6)
Vender	Siemens Healthcare	GE Healthcare	GE Healthcare	Siemens Healthcare	Philips Healthcare
MRI scanner	Skyra 3.0T	Discovery MR750 3.0T	Signa 1.5T EX-HDX	Prisma fit 3.0T	Ingenia CX 3.0T
Sequence	3D-T1-MPRAGE	3D-IR-T1-SPGR	3D-IR-T1-SPGR	3D-T1-MPRAGE	3D-IR-T1-TFE
Repetition time (ms)	1130–2530	8.1	7–22	1570	8.8
Echo time (ms)	1.69–2.52	3.2	2–5	2.2	4.9
Matrix	192–256 × 192–256	256 × 256	256 × 256	256 × 256	288 × 288
FOV	192–220	256	256	256	240

**Table 2 diagnostics-13-02774-t002:** List for label brain volumetric measurements.

ANIMAL Label	Description	ANIMAL Label	Description
2	Rt Parietal GM	53	Rt Caudate
3	Lt Lateral ventricle	57	Lt Parietal WM
4	Rt Occipital GM	59	Rt Temporal WM
6	Lt Parietal GM	67	Lt Cerebellum
8	Lt Occipital GM	73	Lt Occipital WM
9	Rt Lateral ventricle	76	Rt Cerebellum
11	Rt Globus pallidus	83	Lt Temporal WM
12	Lt Globus pallidus	102	Lt Thalamus
14	Lt Putamen	105	Rt Parietal WM
16	Rt Putamen	203	Rt Thalamus
17	Rt Frontal WM	210	Lt Frontal GM
20	Brainstem	211	Rt Frontal GM
23	Rt Subthalamic nucleus	218	Lt Temporal GM
29	Lt Fornix	219	Rt Temporal GM
30	Lt Frontal WM	232	Third ventricle
33	Lt Subthalamic nucleus	233	Fourth ventricle
39	Lt Caudate	254	Rt Fornix
45	Rt Occipital WM	255	Extracerebral CSF
**Global measurements**		**Children (ANIMAL label numbers)**	
Whole brain		2, 3, 4, 6, 8, 9, 11, 12, 14, 16, 17, 20, 23, 29, 30, 33, 39, 45, 53, 57, 59, 67, 73, 76, 83, 102, 105, 203, 210, 211, 218, 219, 232, 233, 254, 255	
Cortical gray matter (CGM)		2, 4, 6, 8, 210, 211, 218, 219	
White matter (WM)		17, 30, 45, 57, 59, 73, 83, 105	
Subcortical gray matter and fornix (SGM)		11, 12, 14, 16, 23, 29, 33, 39, 53, 102, 203, 254	
Extra-axial CSF (CSF)		3, 9, 232, 233, 255	
Cerebellum and brainstem (CB)		20, 67, 76	

**Table 3 diagnostics-13-02774-t003:** Global brain volumetric measurements after harmonization for each age range.

	6 YO ≤ Age < 9 YO							
	Male (*n* = 89)		Female (*n* = 86)		Total (*n* = 175)			
Description	Mean (mm^3^)	SD (mm^3^)	Mean (mm^3^)	SD (mm^3^)	Mean (mm^3^)	SD (mm^3^)	*p*-Value (Male vs. Female)	Absolute Cohen’s *d* (Male vs. Female)
Whole brain *	1,819,754	142,320	1,682,695	143,133	1,752,399	158,031	1.8 × 10^−9^	0.96
Cortical GM *	801,934	63,252	745,662	71,998	774,280	73,155	1.5 × 10^−7^	0.83
WM *	436,705	52,331	390,500	42,780	413,998	53,063	1.4 × 10^−9^	0.97
SGM *	37,714	2708	35,171	2758	36,464	3008	5.2 × 10^−9^	0.93
Extra-axial CSF	363,811	62,982	347,218	62,073	355,657	62,910	0.081	0.27
CB *	179,590	14,080	164,145	13,832	172,000	15,928	9.0 × 10^−12^	1.11
	**9 YO ≤ Age < 12 YO**							
	**Male (*n* = 85)**		**Female (*n* = 127)**		**Total (*n* = 212)**			
Description	Mean (mm^3^)	SD (mm^3^)	Mean (mm^3^)	SD (mm^3^)	Mean (mm^3^)	SD (mm^3^)	*p*-value (male vs. female)	Absolute Cohen’s *d* (male vs. female)
Whole brain *	1,870,988	133,244	1,729,428	140,313	1,786,186	153,820	4.0 × 10^−12^	1.03
Cortical GM *	787,813	63,495	733,343	70,839	755,183	72,922	2.2 × 10^−8^	0.8
WM *	473,186	44,321	424,176	47,318	443,826	51,949	8.5 × 10^−13^	1.06
SGM *	38,328	2530	36,241	3024	37,078	3010	1.6 × 10^−7^	0.74
Extra-axial CSF	388,022	73,526	365,506	69,591	374,534	71,878	0.027	0.32
CB *	183,639	13,842	170,163	14,034	175,566	15,418	7.9 × 10^−11^	0.97
	**12 YO ≤ Age < 15 YO**							
	**Male (*n* = 93)**		**Female (*n* = 115)**		**Total (*n* = 208)**			
Description	Mean (mm^3^)	SD (mm^3^)	Mean (mm^3^)	SD (mm^3^)	Mean (mm^3^)	SD (mm^3^)	*p*-value (male vs. female)	Absolute Cohen’s *d* (male vs. female)
Whole brain *	1,921,579	164,698	1,768,993	116,849	1,837,216	159,244	3.4 × 10^−12^	1.09
Cortical GM *	767,612	76,019	681,028	61,996	719,741	80,915	8.6 × 10^−16^	1.26
WM *	493,138	60,699	443,730	43,184	465,821	57,192	5.3 × 10^−10^	0.95
SGM *	38,053	3318	36,274	2567	37,070	3051	3.6 × 10^−5^	0.61
Extra-axial CSF	435,120	78,133	435,147	66,097	435,135	71,548	0.998	3.8 × 10^−4^
CB *	187,656	14,739	172,814	12,441	179,450	15,379	7.1 × 10^−13^	1.01
	**15 YO ≤ Age < 18 YO**							
	**Male (*n* = 72)**		**Female (*n* = 179)**		**Total (*n* = 251)**			
Description	Mean (mm^3^)	SD (mm^3^)	Mean (mm^3^)	SD (mm^3^)	Mean (mm^3^)	SD (mm^3^)	*p*-value (male vs. female)	Absolute Cohen’s *d* (male vs. female)
Whole brain *	2,014,618	148,885	1,775,840	133,929	1,844,334	175,433	7.8 × 10^−22^	1.73
Cortical GM *	751,846	70,895	645,060	67,774	675,692	83,905	5.8 × 10^−20^	1.55
WM *	528,250	53,938	449,966	50,174	472,422	62,268	4.6 × 10^−19^	1.53
SGM *	39,862	3081	35,853	2632	37,003	3306	1.2 × 10^−16^	1.45
Extra-axial CSF	502,777	80,760	473,363	62,903	481,801	69,622	6.6 × 10^−3^	0.43
CB *	191,883	15,048	171,598	15,155	177,417	17,674	5.7 × 10^−17^	1.34

* Indicates a statistically significant finding based on *p* < 1.4 × 10^−3^ (two-tail unpaired *t*-test). Abbreviations: CB, cerebellum and brainstem; GM, gray matter; CSF, extra-axial cerebrospinal fluid; SGM, Subcortical gray matter and fornix; WM, white matter.

**Table 4 diagnostics-13-02774-t004:** Laterality index for each brain volumetric measurement.

Brain Volumetric Measurements	Male (*n* = 339)		Female (*n* = 507)		Total (*n* = 846)			
	Mean	SD	Mean	SD	Mean	SD	*p*-Value (Male vs. Female)	Absolute Cohen’s *d* (Male vs. Female)
Frontal GM	−0.45	0.91	−0.30	0.98	−0.36	0.96	0.024	0.16
Temporal GM	−0.98	1.58	−0.89	2.04	−0.93	1.87	0.44	0.05
Parietal GM	0.72	1.91	0.66	2.29	0.69	2.15	0.67	0.03
Occipital GM	−1.09	3.29	−0.82	3.27	−0.93	3.28	0.24	0.08
Frontal WM	0.40	1.20	0.37	1.16	0.38	1.18	0.74	0.02
Temporal WM	−0.42	2.04	−0.24	2.13	−0.31	2.10	0.22	0.08
Parietal WM	0.69	2.07	0.62	1.92	0.65	1.98	0.66	0.03
Occipital WM	−0.78	3.84	−0.65	3.62	−0.70	3.71	0.62	0.04
Globus pallidus	1.53	2.16	1.87	2.37	1.73	2.29	0.032	0.15
Putamen	−1.25	1.28	−1.43	1.49	−1.36	1.41	0.051	0.13
Fornix	1.45	2.69	1.65	2.35	1.57	2.49	0.26	0.08
Caudate	0.37	1.92	0.59	1.87	0.50	1.89	0.091	0.12
Thalamus	0.28	0.96	0.36	0.91	0.33	0.93	0.24	0.08
Subthalamic nucleus	−0.12	5.18	0.64	5.09	0.33	5.14	0.035	0.15
Lateral ventricle	2.86	16.94	4.83	16.07	4.04	16.44	0.092	0.12
Cerebellum	0.05	1.18	0.07	0.98	0.06	1.06	0.77	0.02

Abbreviations: GM, gray matter; SD, standard deviation; WM, white matter.

## Data Availability

The participants in this study did not give written consent for their data to be shared publicly, so due to the sensitive nature of the research support, data are not available.

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
