# Peer review of "A Brain Morphometry Study with Across-Site Harmonization Using a ComBat-Generalized Additive Model in Children and Adolescents"

_diagnostics, 2023, doi:10.3390/diagnostics13172774_

Round 1

Reviewer 1 Report

The manuscript under review identifies the normal reference values for global and regional brain measures using data of children and adolescents from different acquisition sites. The authors implemented ComBat-GAM as harmonization model and considered the effect of sex and age group in the analyses.

This work provides a new insight into how brain changes occur in children and adolescents when considering whole-brain and regional volumes. Overall, the paper is well written, and analyses are well conducted. Some issues should be considered:

1)    It would be helpful for the reader to have a definition of what the authors mean by healthy volunteers and neurotypical participants. The two terms seem to be used interchangeably and sometimes create confusion.

2)      How many scans were excluded because of poor quality? Can the authors provide this info broken down by acquisition site?

3)      Given there are analyses comparing male and female volumes, which normalization procedure was performed to account for the overall bigger volumes for males than females?

4)      Should table on page 5 be table 3?

5)      Is the threshold of +/- 20 for the lateralization index arbitrarily set by the authors or based on previous studies? If the latter, a reference is missing.

6)      Page 5, lines 171-173. It is unclear why additional volumes of patients were included, from which site they were recorded from, and the overall rationale for the inclusion. I would suggest adding a better explanation for the inclusion of these volumes.

7)      Please, double-check that all acronyms are spelled out the first time used.  

Reviewer 2 Report

1. Related work was missed

2. Organization of the article needs to include

3. Quality of figures improves with high dpi

4. Add the future scope of the work

Need to improve

Reviewer 3 Report

The article offers the normal reference values between global and regional brain volume sites for each sex and age group in children and adolescents through ComBat-GAM harmonization in 846 patients.

It only uses descriptive data, I wonder if inferential data between sex would be interesting.

1. What is the main question that the research addresses?

Try to create a brain map from different measurements taken in adolescents   2. Do you consider the topic original or relevant in the field? it does address a specific gap in the field?   The subject is relevant, since the brain is an organ that is still very unknown, and can provide information on specific aspects.   3. What does it contribute to the subject area compared to other publications?   It is a study that provides different parameters, useful for comparing with other studies.   4. What specific improvements should authors consider regarding the methodology?   As it is a measurement study, it is necessary to develop inferential analyzes that can allow the differences between sexes to be compared. What additional controls should be considered? Above all, take into account inferential analyses, which help to obtain correlations or mean differences.   5. Are the conclusions consistent with the evidence and arguments presented?   are not consistent, as they are based on descriptive data And do they answer the main question posed? They do not respond, due to lack of appropriate statistics   6. Are the references appropriate?   yes, they are appropriate   7. Include any additional comments about the tables and figures.

Tables and figures are pertinent, as they help to clarify the message.

Round 2

Reviewer 2 Report

1. Figures quality needs to be improve 

2. Tables too length not able to understand the readers
